# DepthFL: Depthwise Federated Learning for Heterogeneous Clients

**Minjae Kim**
Seoul National University
mjkim@snu.ac.kr

**Sangyoon Yu**
Seoul National University
sangyoonyu@snu.ac.kr

**Suhyun Kim**∗
Korea Institute of Science and Technology
dr.suhyun.kim@gmail.com

**Soo-Mook Moon**∗
Seoul National University
smoon@snu.ac.kr

## ABSTRACT

*Federated learning* is for training a global model without collecting private local data from clients. As they repeatedly need to upload locally-updated weights or gradients instead, clients require both computation and communication resources enough to participate in learning, but in reality their resources are heterogeneous. To enable resource-constrained clients to train smaller local models, *width scaling* techniques have been used, which prunes the channels of a global model. Unfortunately, width scaling suffers from parameter mismatches of channels when aggregating them, leading to a lower accuracy than when simply excluding resource-constrained clients from training. This paper proposes a new approach based on *depth scaling* called *DepthFL* to solve this issue. DepthFL defines local models of different depths by pruning the deepest layers off the global model, and allocates them to clients depending on their resources. Since many clients do not have enough resources to train deep local models, this would make deep layers partially-trained with insufficient data, unlike shallow layers that are fully trained. DepthFL alleviates this problem by mutual self-distillation of knowledge among the classifiers of various depths within a local model. Our experiments show that depth-scaled local models build a global model better than width-scaled ones, and that self-distillation is highly effective in training data-insufficient deep layers.

## 1 INTRODUCTION

Federated learning is a type of distributed learning. It trains a shared global model by aggregating locally-updated model parameters without direct access to the data held by clients. It is particularly suitable for training a model with on-device private data, such as next word prediction or on-device item ranking (Bonawitz et al., 2019). Generally, federated learning demands client devices to have enough computing power to train a deep model as well as enough communication resources to exchange the model parameters with the server. However, the computation and communication capability of each client is quite diverse, often changing dynamically depending on its current loads, which can make those clients with the smallest resources become a bottleneck for federated learning. To handle this issue, it would be appropriate for clients to have a different-sized local model depending on their available resources. However, it is unclear how we can create local models of different sizes, without affecting the convergence of the global model and its performance. There are various methods to *prune* a single global model to create heterogeneous local models, such as *HeteroFL* (Diao et al., 2021), *FjORD* (Horváth et al., 2021), and *Split-Mix* (Hong et al., 2022). They create a local model as a subset of the global model by pruning channels, that is, *width*-based scaling. HeteroFL was a cornerstone research that could make different local models by dividing a global model based on width, yet still producing a global model successfully. However, we observed some issues of width scaling. We evaluated HeteroFL compared to *exclusive* federated learning, which

---
∗co-corresponding authors

simply excludes those clients who do not have enough resources to train a given global model from training (see Section 4.2). The result shows that the global model of HeteroFL achieves a tangibly lower accuracy than the models of exclusive learning, due to parameter mismatch of channels when they are aggregated. In this paper, we propose a different approach of making local models, called *DepthFL*. DepthFL divides a global model based on *depth* rather than width. We construct a global model that has several classifiers of different depths. Then, we prune the highest-level layers of the global model to create local models with different depths, thus with a different number of classifiers. We found that this depth-based scaling shows a better performance than exclusive learning in most cases, unlike in HeteroFL, since training a local model can directly supervise its sub-classifiers as well as its output classifier, obviating parameter mismatch of sub-classifiers when aggregated. This means that depth scaling allows resource-constrained clients to participate and contribute to learning, although there is a small overhead of separate classifiers. We analyzed the root cause of this difference between depth scaling and width scaling.

There is one issue in depth scaling, though. There are only a few clients whose local models include deep classifiers, while many clients have shallow classifiers in their local model. This would make deep classifiers be partially-trained only with a limited amount of data, so their accuracy might be inferior to fully-trained shallow classifiers. That is, resource-constrained clients cannot train deep classifiers, so deep classifiers cannot be general enough to cover the unseen data of those clients. To moderate this issue, we make the deep classifiers of a local model learn from its shallow classifiers by knowledge distillation. This is similar to self distillation (Zhang et al., 2019), except that the direction of distillation is opposite. Actually, we make the classifiers collaborate with each other as in *deep mutual learning* (Zhang et al., 2018). So, a client not only trains the classifiers in its local model using its data, but distills each other's knowledge at the same time. Our evaluation shows that deep classifiers can learn from shallow classifiers trained with otherwise unseen data, and that both classifiers actually help each other to improve the overall performance of the global model. We also analyzed the fundamental reason for the effectiveness of knowledge distillation in DepthFL, using an evaluation in a general teacher-student environment.

Recently, InclusiveFL (Liu et al., 2022) proposed a kind of depth-scaled method with a similar intuition to ours, yet with less elaboration. We show that depth scaling alone in InclusiveFL without the companion objective of sub-classifiers cannot effectively solve parameter mismatches, and that its performance of deep classifiers is much lower due to no self distillation. We employed the latest federated learning algorithm *FedDyn* (Acar et al., 2021) as the optimizer for proposed method. Although DepthFL is a framework that focuses on resource heterogeneity of clients, we try to verify if the FedDyn optimizer that focuses on data heterogeneity is applicable to DepthFL seamlessly. Our experiment shows that DepthFL works well for both resource heterogeneity and data heterogeneity.

**Contributions**   To summarize, our contributions are threefold:

- We present a new depth scaling method to create heterogeneous local models with sub-classifiers, which are directly supervised during training, thus no parameter mismatches when aggregated.

- We show that knowledge distillation among the classifiers in a local model can effectively train deep classifiers that can see only a limited amount of data.

- We perform a comprehensive evaluation on the difference between depth scaling and width scaling using exclusive learning, and on the effectiveness of knowledge distillation in DepthFL.

## 2   RELATED WORK

### 2.1   FEDERATED LEARNING

Based on *FedAvg* (McMahan et al., 2017), the standard federated learning method, there have been many studies to solve various problems of federated learning (Li et al., 2020a; Kairouz et al., 2019). One main research field deals with efficient learning algorithms considering the non-IID distribution of data among clients (Karimireddy et al., 2020; Li et al., 2021b; Wang et al., 2020; Acar et al., 2021; Li et al., 2020b). For example, *FedDyn* (Acar et al., 2021) performs exact minimization by making the local objective align with the global objective, which enables fast and stable convergence. There are also studies to deal with heterogeneous, resource-constrained clients in federated learning. One popular method to create heterogeneous local models is to prune the channels of a global model

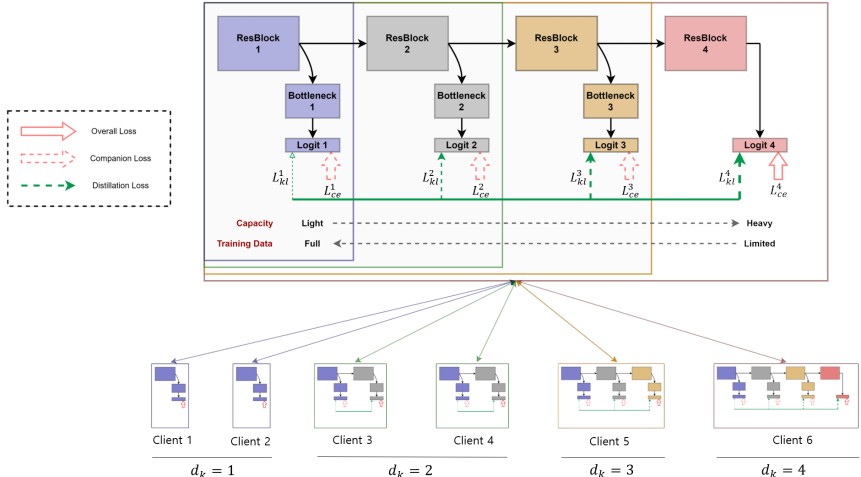

Figure 1: Global model (ResNet) parameter $W_g$ has 4 kinds of local models as its subsets, distributed to $m = 6$ heterogeneous clients. Each local model has multiple classifiers at different depths.

for width scaling (Diao et al., 2021; Li et al., 2021a; Niu et al., 2020; Horváth et al., 2021; Hong et al., 2022; Zhu et al., 2022). Based on the general layer-pruning methods (Chen & Zhao, 2019; Sajjad et al., 2023) that create a shallow model, there is a recent work (Liu et al., 2022) to prune the layers of a global model for tackling system heterogeneity, but it partially exploits the benefit of depth scaling and knowledge distillation unlike DepthFL (see Appendix A.3). Another direction is utilizing knowledge distillation to reduce the burden on clients (He et al., 2020) or to aggregate heterogeneous local models (Zhang & Yuan, 2021; Lin et al., 2020). Separately, there are researches to reduce the communication cost by compressing the local models (Rothchild et al., 2020; Haddadpour et al., 2021; Reisizadeh et al., 2020). There are also research directions on personalized federated learning to handle heterogeneity of clients (Li et al., 2021c; Zhang et al., 2021; Hanzely et al., 2020; T. Dinh et al., 2020).

## 2.2 KNOWLEDGE DISTILLATION

Knowledge distillation (Hinton et al., 2015) was introduced as a way of training a small student model with the help of a big teacher model. It was initially thought that the role of knowledge distillation is transferring high-quality, similarity information among categories, but researches are still on going to understand its impact such as the relationship with label smoothing regularization (Yuan et al., 2020; Tang et al., 2020). As such, variants of knowledge distillation techniques have been proposed (Huang et al., 2021; Park et al., 2019; Zhang et al., 2018; 2019). *Deep mutual learning* (Zhang et al., 2018) allows student models to mutually distill each other's knowledge without a cumbersome teacher model. *Self distillation* (Zhang et al., 2019) distills knowledge within a multi-exit network (Teerapittayanon et al., 2016), from its deep layers to shallow layers for higher performance, or for faster and accurate inference (Phuong & Lampert, 2019). DepthFL also employs self distillation within a local model, yet in an opposite direction, mainly for distilling from fully-trained shallow layers to partially-trained deep layers.

## 3 DEPTHFL METHOD

This section describes our proposed DepthFL method in detail. We assume that a total of $m$ heterogeneous clients participate in federated learning, and that each client $k$ has $d_k$ resource capability. Randomly selected $P_t \in [m]$ clients participate in each communication round.

### 3.1 LOCAL MODELS

We create heterogeneous local models by pruning zero or more of the highest-level layers of the global model, instead of pruning some channels for all layers as in HeteroFL. Each client can con-

tribute to the training of the global model by selecting a local model suitable for its resource amount, in a fixed or dynamic way. Since the highest-level layers are entirely pruned, an additional bottleneck layer is needed for a local model to become an independent classifier. This means that the global model should have a different classifier at a different depth. The structure of a global model that satisfies these conditions can be found in *Deeply-Supervised Nets* (DSN) (Lee et al., 2015). Each classifier shares the blocks that create feature maps, and an additional bottleneck layer is attached to each block so that the local models and the global model have several classifiers inside. Accordingly, the local model parameters of the client whose resource capability is $d_k$ is $W_l^{d_k} = W_g[: d_k]$, and it has $d_k$ classifiers inside. The overall model structure can be found in Figure 1. DSN can provide integrated direct supervision to sub-classifiers in addition to the output classifier. DSN exploits it for better performance, but DepthFL exploits it for consistent training of sub-classifiers across all local models that includes it to fully exhibit its performance.

This way of creating heterogeneous local models is likely to make deep classifiers learn from a partial amount of data, because only resource-rich clients can train local models with deep classifiers. On the other hand, shallow classifiers are likely to be fully trained since many clients can train local models with them. To mitigate the variance of the performance of different classifiers, it is natural to use their ensemble for inference. One interesting question is if partially-trained deep classifiers can really contribute to learning when many clients cannot afford a local model with them. In our experimental result in Table 4, deep classifiers perform similarly or even worse than shallow classifiers unless they receive a help, in the form of *self distillation* described below.

## 3.2 SELF DISTILLATION

To mitigate the low performance of deep classifiers, DepthFL utilizes the concept of self distillation. Several classifiers inside a local model are trained through the cross-entropy loss with labels, as well as the KL loss with the output of other classifiers, as depicted in Figure 1. Previously, self distillation made shallow classifiers imitate deep classifiers (Zhang et al., 2019; Phuong & Lampert, 2019). Whereas, DepthFL makes deep classifiers imitate shallow classifiers, or more precisely, learn collaboratively. The local objective function of the client is as follows:

$$L_k = \sum_{i=1}^{d_k} L_{ce}^i + \frac{1}{d_k - 1} \sum_{i=1}^{d_k} \sum_{j=1, j \neq i}^{d_k} D_{KL}(p_j \| p_i) \tag{1}$$

where $L_{ce}^i$ is a cross-entropy loss of $i$th classifier, and $p_i$ are softmax outputs of $i$th classifier's logits. Although FjORD (Horváth et al., 2021) utilized self distillation already, the motivation of self distillation in FjORD was to allow for the bigger capacity supermodel to teach the width-scaled submodel. In contrast, the purpose of self distillation in DepthFL is that depth-scaled submodels help training of bigger capacity supermodels. Also, unlike distillation between width scaled models, which should run forward pass through teacher supermodel and student submodel independently, there is no overhead of distillation between depth scaled models except for the bottleneck layers because they share feature maps.

We use FedDyn (Acar et al., 2021) instead of FedAvg (McMahan et al., 2017) as the default optimizer for the above local objective function 1, to make a fast convergence even when there is a data heterogeneity of clients. When applying dynamic regularization, we replaced the client's local objective by the modified heterogeneous local objective of 1. Also, $\nabla L_k(\theta_k^t)$ and $h$ values required for dynamic regularization are used and updated in consideration of the heterogeneity of the local models. Therefore, the penalized local objective function of the client is as follows.

$$L_k'(\tilde{\theta}) = L_k(\tilde{\theta}) - \langle \nabla L_k(\tilde{\theta}_k^t), \tilde{\theta} \rangle + \frac{\alpha}{2} \|\tilde{\theta} - \tilde{\theta}^t\|^2 \tag{2}$$

where $\tilde{\theta}$ is the local model parameters, $\nabla L_k(\tilde{\theta}_k^t)$ is the gradient of the local objective function in the previous round, and $\tilde{\theta}^t$ is parts of the current global model's parameters corresponding to the local model parameters $\tilde{\theta}$. Considering the case where the complexity $d_k$ of the client changes dynamically depending on its current loads, $\nabla L_k(\theta_k^t)$ is stored in the same shape as the entire global model parameters, and only a subset corresponding to the current local model parameters is used for actual training. So, the value of $\nabla L_k(\theta_k^t)$ is updated as in the following equation.

$$\nabla L_k(\theta_k^{t+1})[: d_k] \leftarrow \nabla L_k(\theta_k^t)[: d_k] - \alpha(\tilde{\theta}_k^{t+1} - \tilde{\theta}^t) \tag{3}$$

**Algorithm 1:** DepthFL

Initialization : $\theta^0, h^0 = \mathbf{0}, \nabla L_k(\theta_k^0) = \mathbf{0}$

**Server executes:**
**for** *round* $t = 0, 1, \ldots T - 1$ **do**
    $P_t \leftarrow$ Random Clients
    $\theta^{t+1} \leftarrow \mathbf{0}$
    $h^{t+1} \leftarrow h^t$
    **for** *each client* $k \in P_t$, *and in parallel* **do**
        $\tilde{\theta}^t \leftarrow \theta^t[: d_k]$
        $\tilde{\theta}_k^{t+1} \leftarrow$ **Client_Update**$(k, \tilde{\theta}^t)$
        $h^{t+1}[: d_k] \leftarrow h^{t+1}[: d_k] - \alpha \frac{1}{m}(\tilde{\theta}_k^{t+1} - \theta^t[: d_k])$
        $\theta^{t+1}[: d_k] \leftarrow \theta^{t+1}[: d_k] + \tilde{\theta}_k^{t+1}$
    **end**
    **for** *each resource capability* $d_i$ **do**
        $\theta^{t+1}[d_i] = \frac{1}{|P_t^{d_k \geq d_i}|} \theta^{t+1}[d_i] - \frac{1}{\alpha} h^{t+1}[d_i]$
    **end**
**end**

**Client_Update**$(k, \tilde{\theta}^t)$**:**
$\tilde{\theta}_k^{t+1} \leftarrow \tilde{\theta}^t$
$\nabla L_k(\tilde{\theta}_k^t) \leftarrow \nabla L_k(\theta_k^{latest\_updated})[: d_k]$
**for** *local epoch* $e = 1, 2, \ldots E$ **do**
    **for** *each mini batch* $\mathbf{b}$ **do**
        $L_k = \sum\limits_{i=1}^{d_k} L_{ce}^i + \frac{1}{d_k - 1} \sum\limits_{i=1}^{d_k} \sum\limits_{j=1, j \neq i}^{d_k} D_{KL}(p_j \| p_i)$
        $L_k'(\tilde{\theta}) = L_k(\tilde{\theta}) - \langle \nabla L_k(\tilde{\theta}_k^t), \tilde{\theta} \rangle + \frac{\alpha}{2} \|\tilde{\theta} - \tilde{\theta}^t\|^2$
        $\tilde{\theta}_k^{t+1} \leftarrow \tilde{\theta}_k^{t+1} - \eta \nabla L_k'(\tilde{\theta}_k^{t+1}; \mathbf{b})$
    **end**
**end**
$\nabla L_k(\tilde{\theta}_k^{t+1})[: d_k] \leftarrow \nabla L_k(\theta_k^t)[: d_k] - \alpha(\tilde{\theta}_k^{t+1} - \tilde{\theta}^t)$
return $\tilde{\theta}_k^{t+1}$

Table 1: (Number of parameters) / [# of MACs] of local models according to division method

| Model | Method | $a = W_l^1$ | $b = W_l^2$ | $c = W_l^3$ | $d(W_g) = W_l^4$ |
|---|---|---|---|---|---|
| ConvNet | HeteroFL (Width) | 99.0 K [4.11 M] | 391 K [15.3 M] | 877 K [33.6 M] | 1.56 M [58.9 M] |
| | DepthFL (Depth) | 31.4 K [3.46 M] | 178 K [23.6 M] | 750 K [43.7 M] | 1.94 M [62.6 M] |
| Resnet-18 | HeteroFL (Width) | 713 K [35.5 M] | 2.82 M [140 M] | 6.32 M [314 M] | 11.2 M [557 M] |
| | DepthFL (Depth) | 480 K [167 M] | 1.35 M [310 M] | 3.84 M [450 M] | 12.3 M [585 M] |
| Resnet-34 | HeteroFL (Width) | 1.36 M [73.6 M] | 5.38 M [292 M] | 12.1 M [655 M] | 21.4 M [1.16 G] |
| | DepthFL (Depth) | 0.60 M [243 M] | 2.11 M [538 M] | 9.38 M [980 M] | 22.6 M [1.19 G] |

When the local models of different depths are aggregated into a single global model, the local model parameters of the same depths will be aggregated together. This naturally makes shallow layers close to the input aggregate a large number of local model parameters, while deep layers close to the output aggregate a small number of local model parameters. Algorithm 1 shows the full algorithm. Replacing FedDyn optimizer by FedAvg is straightforward.

## 4 EXPERIMENTS

### 4.1 SETTINGS

**Datasets and models** We used MNIST, CIFAR-100, and Tiny ImageNet datasets for the image classification task, and WikiText-2 dataset for the masked language modeling task. CNN composed of 4 convolution layers, Resnet18, and Resnet34 models were used for MNIST, CIFAR-100, and Tiny ImageNet, respectively. The transformer model was used for Wikitext-2. We create four local models with DepthFL, and four local models with HeteroFL by dividing the channels into four equal parts following its division method. Table 1 depicts the model size and number of MACs for the four local models of HeteroFL and DepthFL. For example, the smallest model $a$ has only the classifier $1/4$ in DepthFL, while having $1/4$ channels in HeteroFL. For inference, DepthFL uses the ensemble of all internal classifiers, while HeteroFL uses the global model with all channels.

**Local model** If scaling is done as mentioned above, each depth-scaled local model would have fewer parameters and more MACs than the corresponding width-scaled one, since shallow layers often have fewer parameters yet require more computations. So, for the Resnet global model, the number of MACs decreases linearly as the depth decreases, but the number of model parameters decreases more rapidly, as shown in Table 1 (the global model of DepthFL has slightly more parameters and MACs than that of HeteroFL due to the additional bottleneck layers). This means that the

average communication overhead of depth-scaled clients would be lower than that of width-scaled clients, but it would be opposite for the average computation overhead. Also, depth scaling is less fine-grained than width scaling since depth scaling is more dependent on the layer structure of the global model, and may not be effective if the size of the bottleneck layer is too large. Despite these limitations, depth-scaled local models do not seriously affect the performance of the global model when aggregated, unlike width-scaled local models, as will be explained shortly.

**Default Settings** Unless otherwise stated, the same number of clients are allocated to each of the four different local models (i.e., 25% of the clients are allocated to each of $a$, $b$, $c$, and $d$ local models in Table 1). Also, the data is distributed in an IID manner, *FedDyn* is used as the optimizer, and randomly sampled 10% of the clients among a total of 100 clients participate in each communication round. When the data is distributed in a non-IID manner, as in FedMA (Wang et al., 2020), a *Dirichlet* distribution $\mathbf{p}_c \sim Dir_k(\beta = 0.5)$ was used to allocate $\mathbf{p}_{c,k}$ ratio of data samples belonging to class $c$ to client $k$.

## 4.2 COMPARISON WITH EXCLUSIVE LEARNING

We first evaluate if it is beneficial for resource-constrained clients to participate in learning with depth-scaled local models even if they cannot accommodate a given global model. For this evaluation, we compare the accuracy of DepthFL with the accuracy of its *exclusive learning*, which excludes those clients who do not have enough resources to run a given global model from learning. In DepthFL, $d$ is the global model but we still allow those 75% clients whose resource cannot accommodate $d$ to participate in learning. In exclusive learning, however, if $d$ is the global model, we allow only those 25% clients who can run $d$ to participate in learning. Similarly, if $c$ is the global model, we allow those 50% clients who can run $c$ (including those who can run $d$ but should run $c$ since $c$ is the global model now) to participate in learning. We also evaluate DepthFL without self-distillation and FedDyn (use FedAvg instead), which we call DepthFL(FedAvg), by comparing with its corresponding exclusive learning. Finally, we evaluate HeteroFL by comparing with its corresponding exclusive learning. For a fair comparison, exclusive learning is trained and tested in the same way as the corresponding scaled method. For example, when comparing DepthFL with its exclusive learning, the local objective function of both includes the self distillation and regularization terms. Also, both are tested with the ensemble inference of the internal classifiers. It should be noted that we cannot compare the accuracy of DepthFL and HeteroFL directly, since their local models have a different size, so comparing only with their corresponding exclusive learning is meaningful.

Table 2 shows the results comparing HeteroFL, DepthFL(FedAvg), and DepthFL to their corresponding exclusive learning. In case of HeteroFL, exclusive learning with $b$ as the global model that prunes half of the channels, and with only 75% of the clients participating in learning, shows a tangibly better accuracy than HeteroFL in CIFAR-100. This means that although clients with insufficient resources can participate in learning on the HeteroFL framework, heterogeneous local models appear to deteriorate the global model and produce a worse result. DepthFL(FedAvg) shows a better result than any exclusive learning for MNIST and CIFAR-100, but not for Tiny Imagenet. On the other hand, DepthFL performs better than any of exclusive learning in all datasets. This result indicates that DepthFL can better train the global model using heterogeneous local models.

One question is why HeteroFL shows a tangibly lower performance than exclusive learning, and why this is not the case for DepthFL(FedAvg). To answer these questions, we measured the performance of each *global sub-model* of HeteroFL separately, by running 1/4, 2/4, 3/4, and 4/4 channels of the global model. We also measured the same for InclusiveFL (Liu et al., 2022) without momentum distillation and DepthFL(FedAvg), i.e., the performance of classifiers at 1/4, 2/4, 3/4, and 4/4 of the global model. Table 3 shows the result compared to exclusive learning result of Table 2 (named Classifier 1/4∼4/4). We can see that a global sub-model of HeteroFL mostly performs worse than the corresponding exclusive learning model, while the global sub-models of DepthFL(FedAvg) perform better than exclusive learning. In HeteroFL, for example, only 25% clients train Classifier 1/4 *directly* as in exclusive learning, while the rest 75% clients train it *indirectly* during the training of their 2/4, 3/4, or 4/4 Classifiers, so we suspect that this causes a parameter mismatch for Classifier 1/4 when being averaged, degrading the performance of the global model. To confirm this suspicion, we evaluated a different version of HeteroFL called *SHeteroFL*, proposed in Split-Mix (Hong et al., 2022), where a client trains all affordable sub-models. For example, when a client with Classifier 2/4 trains, it not only trains Classifier 1/4 indirectly when training Classifier 2/4, but trains

Table 2: Accuracy of the global model compared to exclusive learning. 100% ($a$) exclusive learning means, the global model and every local model are equal to $a = W_l^1$ model, and 100% clients participate in learning. Likewise, 25% ($d$) exclusive learning means, the global model and every local model are equal to $d(W_g) = W_l^4$ model, and only 25% clients participate in learning.

| Scaling Method | Dataset | Global Model | Exclusive Learning | | | |
|---|---|---|---|---|---|---|
| | | | 100% ($a$) | 75% ($b$) | 50% ($c$) | 25% ($d$) |
| HeteroFL | MNIST | 99.33 | 99.38 | 99.39 | **99.41** | 99.19 |
| | CIFAR-100 | 56.68 | 63.83 | **65.07** | 61.93 | 52.00 |
| | Tiny ImageNet | 27.22 | **38.99** | 34.68 | 29.96 | 22.43 |
| DepthFL (FedAvg) | MNIST | **99.43** | 99.29 | 99.35 | 99.39 | 99.25 |
| | CIFAR-100 | **72.34** | 66.53 | 69.63 | 68.78 | 60.11 |
| | Tiny ImageNet | 48.02 | 49.69 | **52.55** | 46.05 | 34.40 |
| DepthFL | MNIST | **99.51** | 99.25 | 99.41 | 99.41 | 99.34 |
| | CIFAR-100 | **76.34** | 69.26 | 73.90 | 71.75 | 62.12 |
| | Tiny ImageNet | **60.32** | 52.48 | 59.75 | 55.55 | 42.64 |

Table 3: Accuracy of global sub-models compared to exclusive learning on CIFAR-100.

| Method | Algorithm | Classifier 1/4 | Classifier 2/4 | Classifier 3/4 | Classifier 4/4 |
|---|---|---|---|---|---|
| Width Scaling | Exclusive Learning | **63.83** | 65.07 | 61.93 | 52.00 |
| | HeteroFL | 50.41 | 55.05 | 57.71 | 56.68 |
| | SHeteroFL | 63.57 | **66.14** | **65.44** | **63.98** |
| Depth Scaling | Exclusive Learning | 66.53 | 67.03 | 66.47 | 57.68 |
| | InclusiveFL | 47.25 | 53.36 | 58.57 | 60.10 |
| | DepthFL(FedAvg) | **66.66** | **68.30** | **68.17** | **68.22** |

Classifier 1/4 directly. Table 3 shows that SHeteroFL differs from HeteroFL in that the accuracy of its global sub-models is similar or higher than that of exclusive learning, as in DepthFL. This can also explain the better performance of DepthFL(FedAvg) than its exclusive learning. That is, when a client trains its 2/4 Classifier, it also trains 1/4 Classifier with the companion objective, as explained in Section 3.1. InclusiveFL has no companion objective, showing a worse accuracy in Table 3 (see also Appendix A.3). Actually, DepthFL does not require redundant training of sub-models when training their super-model, unlike SHeteroFL. Moreover, additional training of depth-scaled sub-models can further improve the global model performance, unlike additional training of width-scaled sub-models (see Appendix A.1). DepthFL also works better than Split-Mix for the same amount of MACs (see Appendix A.2). All these results indicate that DepthFL is more effective than other methods for handling parameter mismatches.

### 4.3 ABLATION STUDY

**Knowledge Distillation** To analyze the impact of mutual self-distillation on the performance of the global model, we turn on and off self distillation and measure the accuracy of the global model. We experiment with both IID and non-IID data distribution, whose results are depicted in Table 4. Regardless of data distribution, it shows that self distillation plays a key role in enhancing the accuracy of deep classifiers. That is, deep classifiers tend to perform worse than shallow classifiers when self distillation is off, yet they perform similarly or better in most cases when self distillation is on. Table 4 shows that self distillation improves the accuracy of shallow classifiers as well, which is encouraging because resource-constrained clients should do inference using shallow classifiers.

**Maximum & Dynamic Complexity** We evaluate why self-distillation enhances the accuracy of deep classifiers. Our conjecture was that fully-supervised shallow classifiers can train under-supervised deep classifiers that were partially trained with a limited amount of data. For this evaluation, we constructed a **Maximum** experimental environment where all clients can train the full global model (i.e., the $d$ model in Table 1). We performed mutual self-distillation on the Maximum environment and measure the accuracy as in Table 5, which includes the previous IID result of Table 4 (marked **Fixed**) for comparison. We can actually see that the impact of self-distillation on deep

Table 4: Accuracy of the global model with/without self distillation for both IID/Non-IID data

| Distribution | Dataset | KD | Classifier 1/4 | Classifier 2/4 | Classifier 3/4 | Classifier 4/4 | Ensemble |
|---|---|---|---|---|---|---|---|
| IID | CIFAR-100 | ✗ | 69.38 | 69.53 | 69.00 | 68.91 | 74.55 |
| | | ✓ | 71.68 (**+2.30**) | 73.89 (**+4.36**) | 73.72 (**+4.72**) | 73.56 (**+4.65**) | 76.34 (**+1.79**) |
| | Tiny ImageNet | ✗ | 51.92 | 52.76 | 41.17 | 40.91 | 52.50 |
| | | ✓ | 55.66 (**+3.74**) | 58.50 (**+5.74**) | 54.83 (**+13.66**) | 55.20 (**+14.29**) | 60.32 (**+7.82**) |
| Non-IID | CIFAR-100 | ✗ | 66.68 | 67.68 | 67.66 | 67.25 | 72.64 |
| | | ✓ | 69.61 (**+2.93**) | 71.46 (**+3.78**) | 71.52 (**+3.86**) | 71.36 (**+4.11**) | 74.25 (**+1.61**) |
| | Tiny ImageNet | ✗ | 50.33 | 50.76 | 44.11 | 43.61 | 53.14 |
| | | ✓ | 53.84 (**+3.51**) | 56.13 (**+5.37**) | 54.28 (**+10.17**) | 53.97 (**+10.36**) | 58.99 (**+5.85**) |

Table 5: Ablation study of the self distillation according to the resource complexity $d_k$ distribution of the clients. Fixed complexity means that a client's complexity $d_k$ does not change from the initial value. Dynamic complexity means a client's $d_k$ value changes randomly every round. Maximum is the situation when all clients have sufficient resources, so all $d_k$ values are maximum. It also shows that DepthFL with self-distillation works well for the transformer model with WikiText-2.

| Dataset | Complexity | KD | Classifier 1/4 | Classifier 2/4 | Classifier 3/4 | Classifier 4/4 | Ensemble |
|---|---|---|---|---|---|---|---|
| CIFAR100 | Fixed | ✗ | 69.38 | 69.53 | 69.00 | 68.91 | 74.55 |
| | | ✓ | 71.68 (**+2.30**) | 73.89 (**+4.36**) | 73.72 (**+4.72**) | 73.56 (**+4.65**) | 76.34 (**+1.79**) |
| | Dynamic | ✗ | 69.33 | 72.44 | 73.97 | 73.92 | 77.14 |
| | | ✓ | 72.87(**+3.54**) | 75.21(**+2.77**) | 75.42(**+1.45**) | 75.51(**+1.59**) | 77.49(**+0.35**) |
| | Maximum | ✗ | 68.99 | 72.36 | 74.42 | 74.37 | 77.48 |
| | | ✓ | 72.29(**+3.30**) | 75.04(**+2.68**) | 76.47(**+2.05**) | 76.54(**+2.17**) | 78.02(**+0.54**) |
| Tiny ImageNet | Fixed | ✗ | 51.92 | 52.76 | 41.17 | 40.91 | 52.50 |
| | | ✓ | 55.66 (**+3.74**) | 58.50 (**+5.74**) | 54.83 (**+13.66**) | 55.20 (**+14.29**) | 60.32 (**+7.82**) |
| | Dynamic | ✗ | 52.20 | 56.76 | 54.79 | 54.96 | 60.49 |
| | | ✓ | 56.10 (**+3.90**) | 60.35 (**+3.59**) | 60.02 (**+5.23**) | 60.09 (**+5.13**) | 62.71 (**+2.22**) |
| | Maximum | ✗ | 51.36 | 57.30 | 53.27 | 54.05 | 59.88 |
| | | ✓ | 55.83 (**+4.47**) | 60.62 (**+3.32**) | 59.88 (**+6.61**) | 60.40 (**+6.35**) | 63.47 (**+3.59**) |
| WikiText-2 (Perplexity ↓) | Fixed | ✗ | 13.21 | 13.42 | 13.48 | 14.31 | 13.08 |
| | | ✓ | 12.99 (**-0.22**) | 13.24 (**-0.18**) | 13.38 (**-0.10**) | 13.99 (**-0.32**) | 13.06 (**-0.02**) |
| | Dynamic | ✗ | 13.21 | 13.31 | 13.38 | 13.80 | 13.02 |
| | | ✓ | 12.97 (**-0.24**) | 13.21 (**-0.10**) | 13.29 (**-0.09**) | 13.71 (**-0.09**) | 13.03 (**+0.01**) |
| | Maximum | ✗ | 13.28 | 13.22 | 12.97 | 12.87 | 12.97 |
| | | ✓ | 13.16 (**-0.12**) | 13.16 (**-0.06**) | 12.91 (**-0.06**) | 12.86 (**-0.01**) | 12.94 (**-0.03**) |

classifiers in Maximum is tangibly smaller than in Fixed. This means that when resources are heterogeneous, self-distillation achieves additional performance by transmitting the domain knowledge of shallow classifiers to deep classifiers. We also constructed a **Dynamic** experimental environment where the resources of the clients can change randomly in every round, yet the average amount of resources of those clients participating in each round is the same as in Fixed. So, unlike Fixed, deeper layers in Dynamic can learn from all data from every client as in Maximum, albeit less frequently. In Table 5, we can see that the impact of self-distillation on deep classifiers in Dynamic is similar to Maximum. This indicates that the lower accuracy of deep classifiers in Fixed is indeed due to those unseen data of resource-constrained clients, and that self-distillation can alleviate the problem.

## 4.4 UNDERSTANDING SELF-DISTILLATION EFFECT OF DEPTHFL

This section attempts to understand the fundamental reason behind the effectiveness of self-distillation for depthFL. For this we evaluate knowledge distillation in a general setup of teacher-student models in a centralized learning environment, except that the student model is trained with insufficient data (as the partially-trained, deep layers of DepthFL). We first evaluate if the existing analysis methods of knowledge distillation in (Tang et al., 2020), which are *label smoothing* (LS), *gradient rescaling* (KD-pt), and *domain knowledge of class relationships* (KD-sim), can explain the effectiveness of self-distillation in DepthFL (see A.6 for detailed explanation). The Resnet18 model is used as the student, while the Resnet101 model, which is fully trained with all data, is used as the teacher. Additionally, we analyze the effect of knowledge distillation by the *poor teacher* (PKD), which is trained only with 25% of the data, as well as by the *light teacher* (LKD), which

Table 6: Top-1 accuracy of student on CIFAR-100 with a few partial knowledge distillation methods.

| Data Amount Seen by Student | Base | KD (Acc: 79.99) | LS | KD-pt (Tang et al., 2020) | KD-sim (Tang et al., 2020) | KD-pt+sim (Tang et al., 2020) | PKD (Acc: 58.14) | LKD (Acc: 72.19) |
|---|---|---|---|---|---|---|---|---|
| 100% | 77.60 | 79.47 | 78.52 | 78.85 | 78.50 | 78.80 | 72.12 | 77.18 |
| 50% | 70.40 | 74.22 | 71.36 | 71.59 | 71.18 | 72.18 | 67.98 | 75.10 |
| 25% | 59.10 | 66.26 | 60.31 | 61.11 | 61.90 | 62.05 | 63.30 | 71.46 |

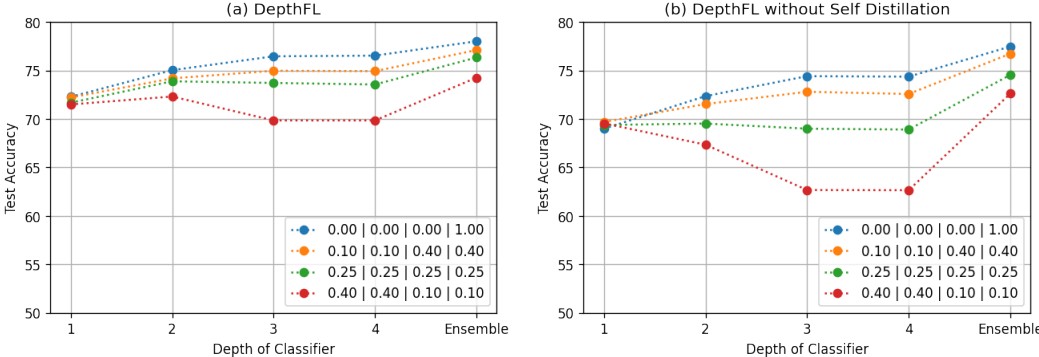

Figure 2: Top-1 accuracy of four classifiers for four resource distribution ratios on CIFAR-100.

is trained with all data but is sized smaller than the student model (as the fully-trained, shallow layers of DepthFL). As the light teacher of LKD, we use $\times 0.25$ depth-scaled local model $a = W_l^1$ of Resnet18. We perform these experiments with a variable amount of data the student learns as in Table 6. When the student model learns all data (100%), most effects of knowledge distillation could be explained by the three existing effects. However, when the student model learns partly (50% or 25%), the effect is not fully explainable by the composition of those three effects, neither by PKD. On the other hand, LKD is quite effective, achieving high accuracy even for 50% and 25%. This means that even if the size of the light teacher is smaller than that of the student, the class relationship for each input instance can be transferred by the light teacher, which effectively helps for the generalization of the data-insufficient student model. This additional role appears to be the reason why self-distillation in DepthFL is effective, especially for data-insufficient deep layers.

## 4.5 ROBUSTNESS TEST

The performance of DepthFL would inevitably be different depending on the distribution of each client's resource capability. To evaluate the robustness of DepthFL, we changed the distribution of resource capabilities and measured the performance of the classifiers, as in Figure 2. As expected, as the ratio of resource-constrained clients increases, the performance of deep classifiers gets lower, even seriously when self distillation is off. With self distillation on, however, even if the deepest classifier is trained only by 10% of the clients, the performance drop is small, due to the help of other classifiers. Since shallow classifiers will be used for fast inference while an ensemble model will be used for high performance, deep classifiers do not always have to be better than shallow ones. Appendix A.5.1 explains and evaluates whether deep classifiers are really needed for DepthFL.

## 5 CONCLUSION

We presented DepthFL, a new federated learning framework considering resource heterogeneity of clients. DepthFL creates local models of different sizes by scaling the depth of the global model, and allocates them to the clients depending on their available resources. During local training, a client trains several classifiers within its local model, and at the same time, distills their knowledge with each other. As a result, both deep classifiers trained with limited data and shallow classifiers trained by most clients can help one another to build the global model, with no parameter mismatch. We also evaluated depth scaling compared to width scaling thoroughly, and self-distillation in DepthFL.

ACKNOWLEDGMENTS

This work was supported by Institute of Information & communications Technology Planning & Evaluation (IITP) grant funded by the Korea government (MSIT) (No. 2021-0-00180, 25%) and (No. 2021-0-00136, 25%), by the ITRC (Information Technology Research Center) support program (IITP-2021-0-01835, 25%) supervised by the IITP, and by the National Research Foundation of Korea (NRF) grant funded by the Korea government (MSIT) (No. RS-2023-00208245, 25%).

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

# A APPENDIX

## A.1 LIMITATION OF SHETEROFL

As mentioned in Section 4.2, although SHeteroFL includes additional computation overhead of locally training all possible sub-models, the performance of its global model that leverages all these sub-models can improve. However, we want to check whether additional training of sub-models of different widths is always helpful for the global model. We compared the performance of FedAvg, SHeteroFL, and DepthFL(FedAvg) in the **Maximum** case introduced in Section 4.3 where all clients have enough resources to train the global model. Table 7 shows the experimental result on CIFAR-100 using Resnet18 as the global model. SHeteroFL does not show better performance than FedAvg despite its additional training of width-scaled sub-models. On the other hand, DepthFL could improve the performance of the global model by additionally training depth-scaled sub-models, which is in line with the result in Lee et al. (2015). In other words, although SHeteroFL could alleviate the problem of HeteroFL, DepthFL is a better approach to heterogeneous federated learning since it can better enhance the performance of global model with less computation cost.

Table 7: Accuracy of global sub-models in **Maximum** case.

| Method | Classifier 1/4 | Classifier 2/4 | Classifier 3/4 | Classifier 4/4 |
|---|---|---|---|---|
| FedAvg | N/A | N/A | N/A | 70.59 |
| SHeteroFL | 62.95 | 68.43 | 69.66 | 70.16 |
| DepthFL (FedAvg) | 66.65 | 70.11 | 73.15 | **73.36** |

## A.2 COMPARISON WITH SPLIT-MIX (HONG ET AL., 2022)

Split-Mix provides a customizable global model by *mixing* the base *split* models of the same sizes. While HeteroFL and DepthFL prune a single global model to create local models of different sizes, Split-Mix can create a local model by combining small base models. To compare Split-Mix with DepthFL, we combine its base models to create four local models whose number of MACs is the same as that of the four local models of DepthFL. For the CIFAR-100 dataset, Resnet18 was used as a global model. The number of MACs and the number of parameters of four local models are shown in Table 8. As the base model of Split-Mix, half of the channels of the global model were pruned. For a fair comparison, DepthFL was trained without regularization and self distillation.

Table 8: (Number of parameters) / [# of MACs] of local models according to division method

| Model | Method | $a = W_l^1$ | $b = W_l^2$ | $c = W_l^3$ | $d(W_g) = W_l^4$ |
|---|---|---|---|---|---|
| Resnet-18 | DepthFL (Depth) | 480 K [167 M] | 1.35 M [310 M] | 3.84 M [450 M] | 12.3 M [585 M] |
| | Split-Mix (Width) | 2.82 M [140 M] | 5.64 M [280 M] | 8.46 M [420 M] | 11.28 M [560 M] |

The accuracy of DethpFL and Split-Mix in **Fixed** and **Dynamic** cases are shown in Figure 3. In Fixed, the learning curves of DepthFL and Split-Mix are almost same. Since the number of parameters of local models of DepthFL is much smaller than Split-Mix, DepthFL is more efficient in terms of communication overhead. In Dynamic, DepthFL shows faster convergence and higher accuracy than Split-Mix. The reason is that Split-Mix allows the client to train multiple base models alternately, so there is no big difference between Fixed and Dynamic, whereas in DepthFL, the deep classifier can train more data in Dynamic, as explained in Section 4.3.

## A.3 COMPARISON WITH INCLUSIVEFL (LIU ET AL., 2022)

Recently, InclusiveFL (Liu et al., 2022) proposed a kind of depth-scaled method for heterogeneous federated learning. However, there are two major differences from DepthFL. First, unlike DepthFL, InclusiveFL trains only the last classifier of the local model without training the sub-models. As we saw in Table 3, training sub-models during local training has a decisive impact on the performance of global sub-models, hence the overall performance (its ensemble result is 63.79, lower than the

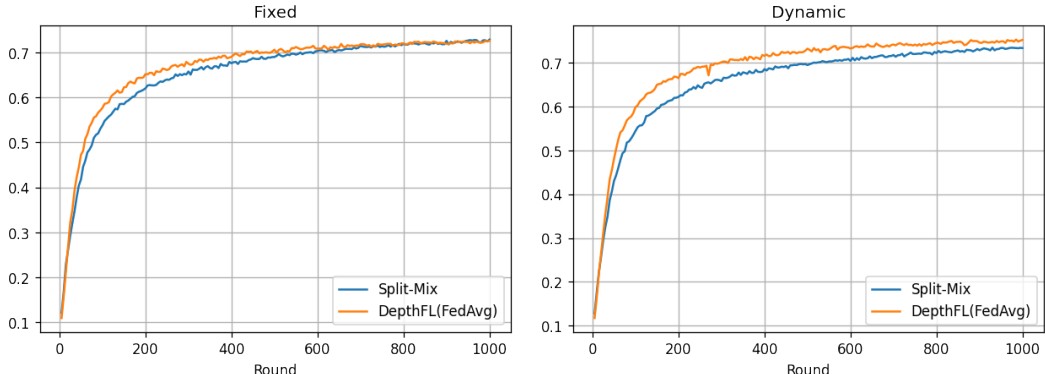

Figure 3: Learning curves for DepthFL and Split-Mix

ensemble result 72.34 of DepthFL). That is, similar to the comparison result between SHeteroFL and HeteroFL, DepthFL has superior performance of sub-classifiers compared to InclusiveFL, and this has a direct impact on the performance of the global model. Actually, unlike DepthFL, InclusiveFL uses only the output of the last classifier during inference, so the performance of the sub classifiers is not that important, and InclusiveFL has no overhead for bottleneck layers. Nevertheless, the performance of Classifier 4/4 in InclusiveFL is even lower than exclusive learning 1/4~3/4, which indicates that training all sub-models in the local model using the companion objective is critical.

Second, since InclusiveFL locally trains only the last classifier in the local model, knowledge distillation between sub-models is impossible. InclusiveFL uses a special technique called *momentum distillation* to transfer the knowledge of a larger model to a smaller model, which has a limitation since the size of each layer must be kept constant. Also, its role is different from self-distillation in DepthFL, which transfers knowledge of a smaller model to a larger model. In summary, since InclusiveFL does not train sub-models in the local model through companion objective nor employ self distillation, InclusiveFL does not fully exploit the advantages of depth scaling unlike DepthFL.

### A.4 OPTIMIZER

DepthFL employs FedDyn (Acar et al., 2021) as the default optimizer. To analyze the impact of the optimizer, we replaced FedDyn with FedAvg (McMahan et al., 2017). And, we performed the same experiments in Table 4 and Table 5, whose results are shown in Table 9 and Table 10, respectively. Even if FedAvg is used as the optimizer, the overall trend of the experimental results is almost the same, but we can see that the overall accuracy is lowered. Also, we can see that the impact of self-distillation is reduced compared to when FedDyn is used.

Table 9: Accuracy of FedAvg global model with/without self distillation for both IID/Non-IID data

| Distribution | Dataset | KD | Classifier 1/4 | Classifier 2/4 | Classifier 3/4 | Classifier 4/4 | Ensemble |
|---|---|---|---|---|---|---|---|
| IID | CIFAR-100 | ✗ | 66.66 | 68.30 | 68.17 | 68.22 | 72.34 |
| | | ✓ | 68.55 (**+1.89**) | 70.23 (**+1.93**) | 71.21 (**+3.04**) | 71.10 (**+2.88**) | 72.96 (**+0.62**) |
| | Tiny ImageNet | ✗ | 49.38 | 48.76 | 38.18 | 38.44 | 48.02 |
| | | ✓ | 51.29 (**+1.91**) | 52.56 (**+3.80**) | 48.99 (**+10.81**) | 49.66 (**+11.22**) | 53.76 (**+5.74**) |
| Non-IID | CIFAR-100 | ✗ | 65.82 | 66.22 | 66.22 | 65.65 | 70.63 |
| | | ✓ | 67.14 (**+1.32**) | 68.41 (**+2.19**) | 69.38 (**+3.16**) | 68.84 (**+3.19**) | 71.48 (**+0.85**) |
| | Tiny ImageNet | ✗ | 47.01 | 46.55 | 39.36 | 38.74 | 47.67 |
| | | ✓ | 49.05 (**+2.04**) | 49.73 (**+3.18**) | 47.03 (**+7.67**) | 46.86 (**+8.12**) | 51.67 (**+4.00**) |

Table 10: Ablation study of self-distillation according to resource complexity $d_k$ distribution.

| Dataset | Complexity | KD | Classifier 1/4 | Classifier 2/4 | Classifier 3/4 | Classifier 4/4 | Ensemble |
|---|---|---|---|---|---|---|---|
| CIFAR100 | Fixed | ✗ | 66.66 | 68.30 | 68.17 | 68.22 | 72.34 |
| | | ✓ | 68.55 (**+1.89**) | 70.23 (**+1.93**) | 71.21 (**+3.04**) | 71.10 (**+2.88**) | 72.96 (**+0.62**) |
| | Dynamic | ✗ | 67.09 | 70.94 | 73.32 | 73.48 | 75.46 |
| | | ✓ | 69.18(**+2.09**) | 72.51(**+1.57**) | 74.28(**+0.96**) | 74.34(**+0.86**) | 75.04(**-0.42**) |
| | Maximum | ✗ | 66.65 | 70.11 | 73.15 | 73.36 | 75.05 |
| | | ✓ | 69.07(**+2.42**) | 71.92(**+1.81**) | 74.77(**+1.62**) | 75.09(**+1.73**) | 75.41(**+0.36**) |
| Tiny ImageNet | Fixed | ✗ | 49.38 | 48.76 | 38.18 | 38.44 | 48.02 |
| | | ✓ | 51.29 (**+1.91**) | 52.56 (**+3.80**) | 48.99 (**+10.81**) | 49.66 (**+11.22**) | 53.76 (**+5.74**) |
| | Dynamic | ✗ | 49.41 | 53.29 | 48.76 | 49.29 | 54.81 |
| | | ✓ | 52.06 (**+2.65**) | 55.25 (**+1.96**) | 54.79 (**+6.03**) | 55.08 (**+5.79**) | 57.40 (**+2.59**) |
| | Maximum | ✗ | 48.58 | 52.82 | 46.97 | 48.06 | 53.77 |
| | | ✓ | 51.45 (**+2.87**) | 55.09 (**+2.27**) | 52.98 (**+6.01**) | 53.97 (**+5.91**) | 56.84 (**+3.07**) |
| WikiText-2 (Perplexity ↓) | Fixed | ✗ | 14.28 | 14.47 | 14.92 | 15.64 | 14.19 |
| | | ✓ | 14.26 (**-0.02**) | 14.44 (**-0.03**) | 14.85 (**-0.07**) | 15.47 (**-0.17**) | 14.21 (**+0.02**) |
| | Dynamic | ✗ | 14.27 | 14.40 | 14.85 | 15.26 | 14.10 |
| | | ✓ | 14.23 (**-0.04**) | 14.39 (**-0.01**) | 14.80 (**-0.05**) | 15.19 (**-0.07**) | 14.14 (**+0.04**) |
| | Maximum | ✗ | 14.25 | 14.32 | 14.67 | 14.91 | 13.96 |
| | | ✓ | 14.23 (**-0.02**) | 14.37 (**+0.05**) | 14.69 (**+0.02**) | 14.92 (**+0.01**) | 14.06 (**+0.10**) |

## A.5   MORE ABLATION STUDIES

### A.5.1   DEEP CLASSIFIERS ARE REALLY NEEDED?

Table 4 showed that the performance of the deepest classifiers is similar or slightly lower than that of shallow classifiers even with self-distillation. So, one might question if deep classifiers are really needed for resource-heterogeneous federated learning, since they are likely to have fewer clients to train although they allow more generalization of the model. That is, if most clients have insufficient resources, deeper layers might not help; otherwise, they might contribute to enhancing the performance of the global model. On the other hand, as in the **Dynamic case** in Table 5, when the client resources change dynamically over time so that all clients can train deep layers even occasionally, deep layers can have a higher performance.

In reality, the performance of classifiers with different depths would inevitably be different depending on various factors such as the distribution of client resources, the nature of the learning task, the number of data each client has, and the size of global model/sub-models. We tried to test whether deep classifiers are really necessary by performing an experiment excluding deep classifiers, whose result is in Figure 4. For example, DepthFL (exclude #4 Classifier) reduces the depth of the global model to 3/4, so that those 25% clients whose resource can train Classifier 4/4 trains only up to Classifier 3/4. We conducted the experiment with big and small global models (Resnet18 and ConvNet) for two learning environments (Fixed and Dynamic) on CIFAR-100. We can see that if the size of the global model is smaller compared to the task, and if the client resources change dynamically so the deep classifier can see more data, deep classifiers gets more important for high performance. Actually, this paper presented a general framework that can flexibly deal with such diverse situations.

### A.5.2   NON-IID DEGREE AND NUMBER OF CLIENTS

We performed ablation study for diverse non-IID degrees and for different number of clients, whose results are in Table 11 and Table 12, respectively. They show a similar behavior as previously.

## A.6   PARTIAL KNOWLEDGE DISTILLATION METHODS

The experiment in Section 4.4 was conducted as follows. In KD-pt method, we synthesize teacher distribution $\rho^{pt}$ as $\rho_i^{pt} = p_t$ if $i = t$, $(1 - p_t)/(K - 1)$ otherwise, where $p_t$ is prediction on ground truth class from the teacher's probability distribution. In KD-sim method, we synthesize teacher distribution $\rho^{sim}$ as the softmax over cosine similarity between the teacher's last logit layer's weights: $\rho^{sim} = \text{softmax}(\text{relu}(\hat{w}_t \hat{W}^T)^\alpha/\beta)$, where $\hat{W} \in \mathbb{R}^{K \times d}$ is the normalized logit layer

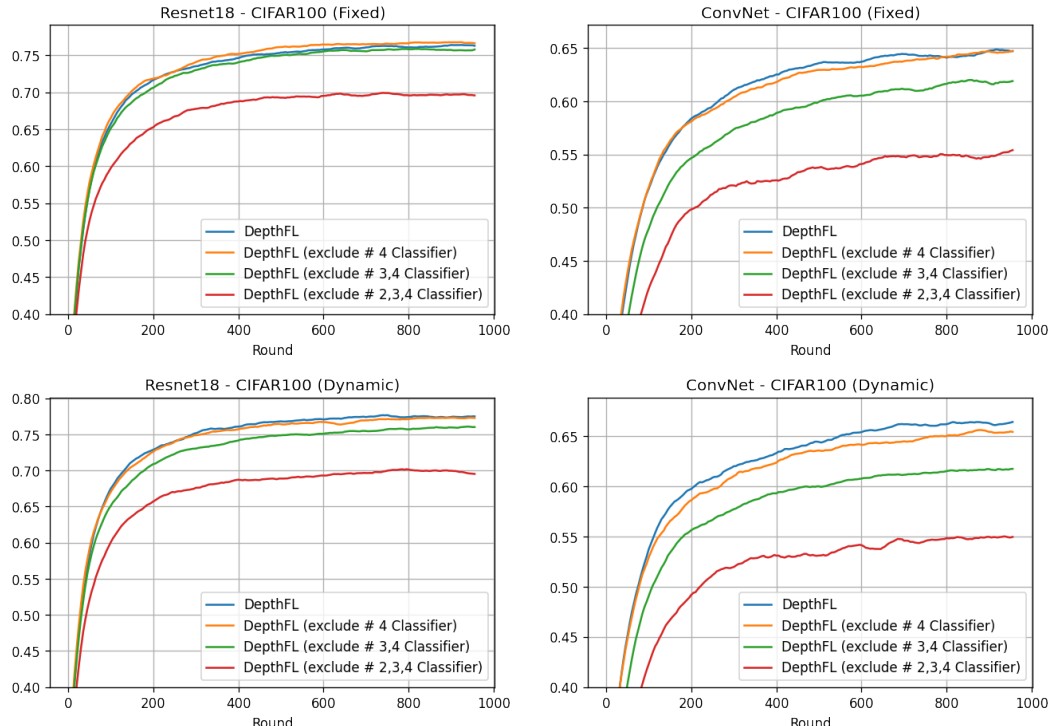

Figure 4: Learning curves for DepthFL without deep classifiers

Table 11: Ablation Study : Data distribution

| Dataset | Distribution | KD | Classifier 1/4 | Classifier 2/4 | Classifier 3/4 | Classifier 4/4 | Ensemble |
|---|---|---|---|---|---|---|---|
| CIFAR-100 | IID | ✗ | 69.38 | 69.53 | 69.00 | 68.91 | 74.55 |
| | | ✓ | 71.68 (**+2.30**) | 73.89 (**+4.36**) | 73.72 (**+4.72**) | 73.56 (**+4.65**) | 76.34 (**+1.79**) |
| | Dir(0.5) | ✗ | 66.68 | 67.68 | 67.66 | 67.25 | 72.64 |
| | | ✓ | 69.61 (**+2.93**) | 71.46 (**+3.78**) | 71.52 (**+3.86**) | 71.36 (**+4.11**) | 74.25 (**+1.61**) |
| | Dir(0.3) | ✗ | 65.69 | 66.83 | 66.68 | 66.24 | 71.54 |
| | | ✓ | 68.46 (**+2.77**) | 70.82 (**+3.99**) | 70.92 (**+4.24**) | 70.57 (**+4.33**) | 73.51 (**+1.97**) |
| | Dir(0.1) | ✗ | 60.97 | 62.35 | 61.97 | 60.79 | 67.35 |
| | | ✓ | 63.88 (**+2.91**) | 65.06 (**+2.71**) | 64.93 (**+2.96**) | 64.14 (**+3.35**) | 68.24 (**+0.89**) |

Table 12: Ablation Study : # Clients

| Dataset | # Clients | KD | Classifier 1/4 | Classifier 2/4 | Classifier 3/4 | Classifier 4/4 | Ensemble |
|---|---|---|---|---|---|---|---|
| CIFAR-100 | 50 | ✗ | 69.20 | 68.95 | 68.53 | 68.59 | 74.39 |
| | | ✓ | 71.88 (**+2.68**) | 72.74 (**+3.79**) | 72.96 (**+4.43**) | 73.02 (**+4.43**) | 76.12 (**+1.73**) |
| | 100 | ✗ | 69.38 | 69.53 | 69.00 | 68.91 | 74.55 |
| | | ✓ | 71.68 (**+2.30**) | 73.89 (**+4.36**) | 73.72 (**+4.72**) | 73.56 (**+4.65**) | 76.34 (**+1.79**) |
| | 200 | ✗ | 68.59 | 69.18 | 68.57 | 68.88 | 73.89 |
| | | ✓ | 71.82 (**+3.23**) | 72.58 (**+3.40**) | 73.37 (**+4.80**) | 73.23 (**+4.35**) | 75.47 (**+1.58**) |

weights, $\hat{w}_t$ is the $t$-th row of $\hat{W}$ corresponding to the ground truth, and $\alpha, \beta$ are hyper-parameters for resolution of cosine similarities.

## A.7 LEARNING CURVES

The learning curves of Table 2 for comparing HeteroFL, DepthFL(FedAvg), and DepthFL with their corresponding exclusive learning are in Figure 5. The learning curves of Table 4 for the ablation study of knowledge distillation are in Figure 6. The x-axis is the communication round, and the y-axis is the moving average of the test accuracy.

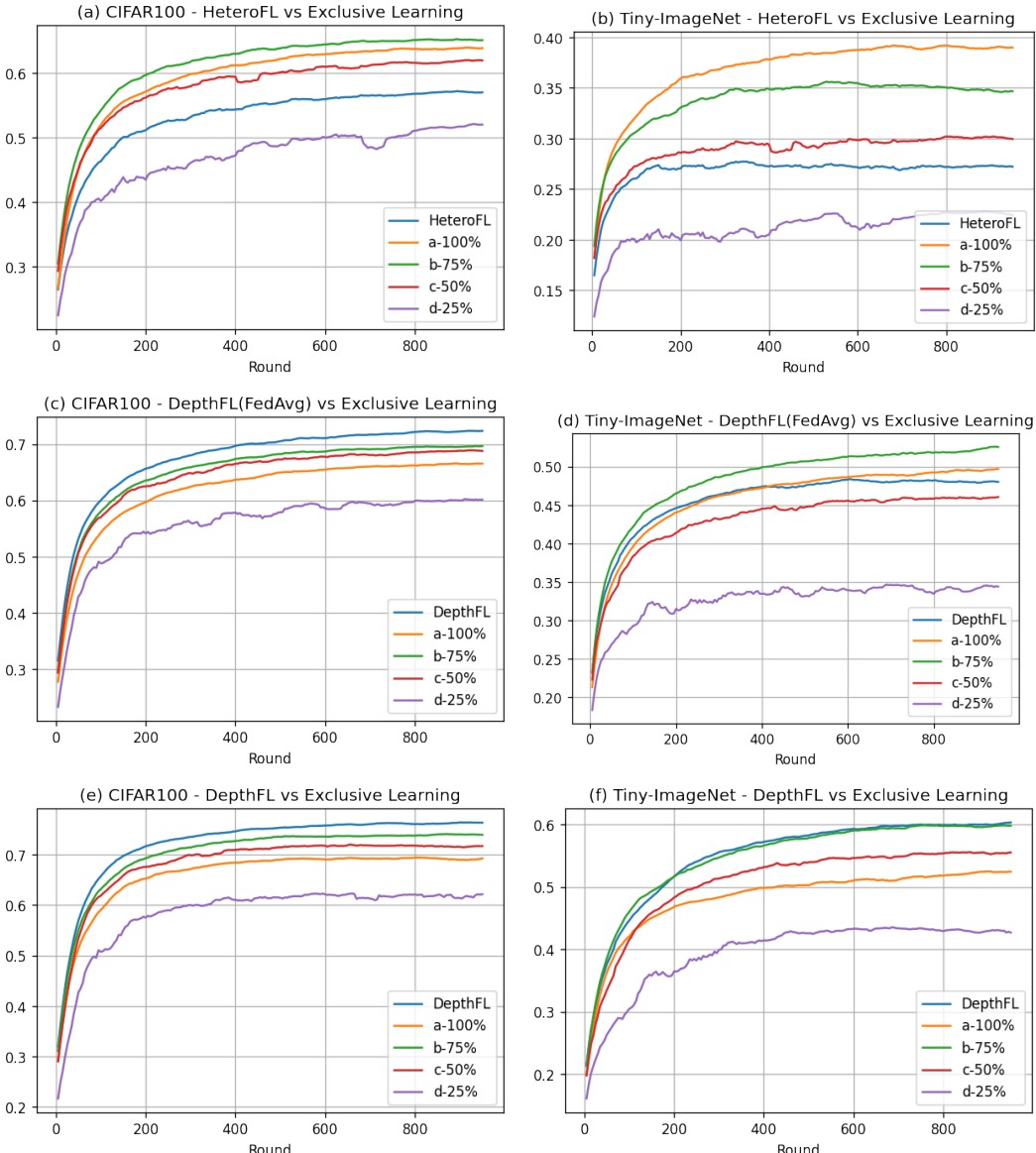

Figure 5: Comparative experiments with exclusive learning

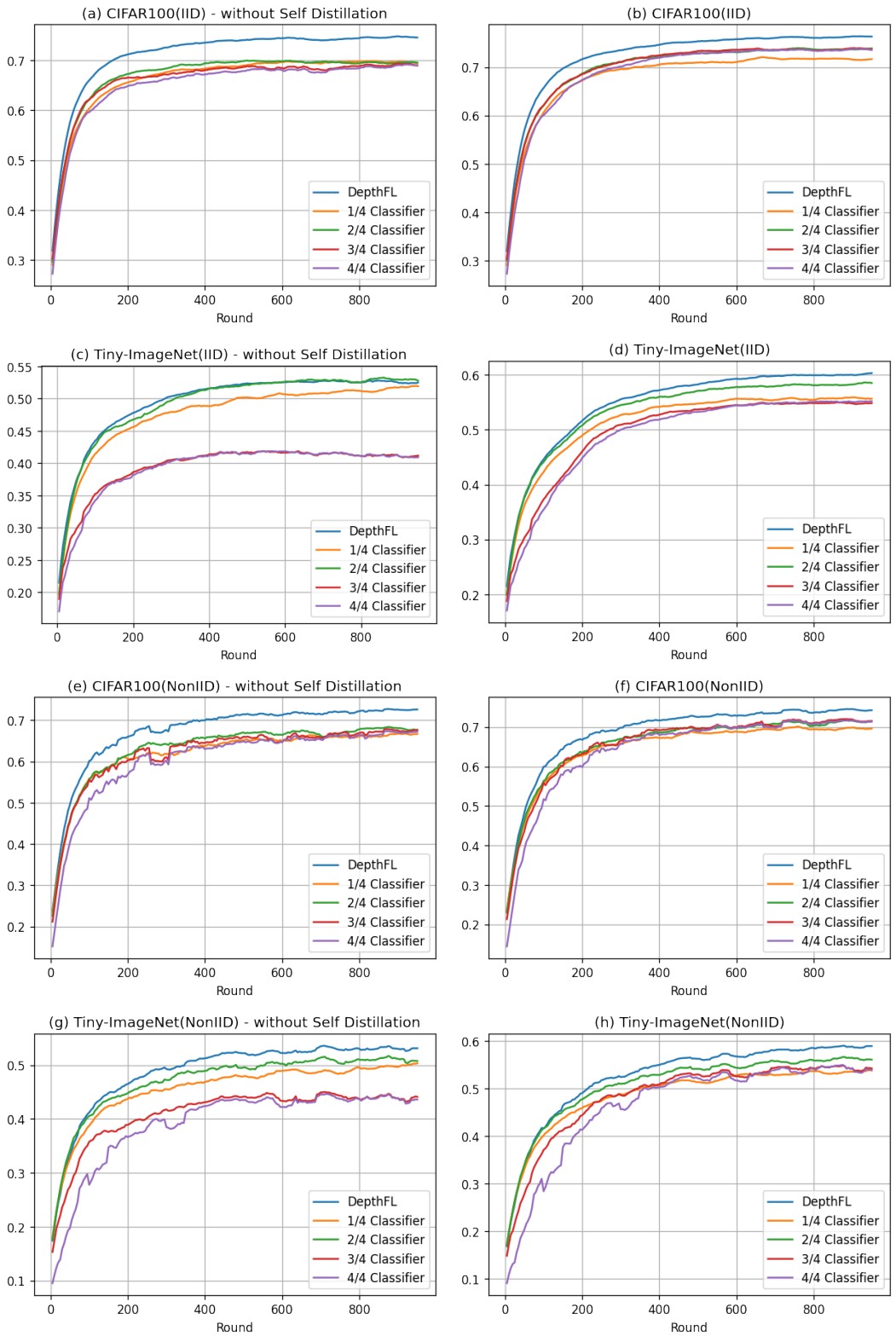

Figure 6: Ablation study of self distillation

## B  HYPERPARAMETERS

Most of the hyperparameters used in our experiments are depicted in Table 13. We set both $\alpha, \beta$ hyper-parameters to 0.3 for KD-sim (Tang et al., 2020) experiments. In the masked language modeling task with the transformer model, we randomly selected 15% of the input tokens. Among the selected tokens, 80% were modified to [mask] tokens, 10% to random tokens, and 10% to remain unchanged.

Table 13: Hyperparameters and model architecture used in experiments

| Data | MNIST | CIFAR100 | Tiny-ImageNet | WikiText-2 |
|---|---|---|---|---|
| Model | ConvNet | Resnet-18 | Resnet-34 | Transformer |
| Hidden size | [64, 128, 256, 512] | [64, 128, 256, 512] | [64, 128, 256, 512] | [512, 512, 512, 512] |
| Local Epoch $E$ | 5 | 5 | 5 | 1 |
| Local Batch Size $B$ | 64 | 64 | 64 | 50 |
| Optimizer | | SGD | | |
| Momentum | | 0. | | |
| Weight decay | | 1e-3 | | |
| Temperature | | 1 | | |
| $\alpha$ (FedDyn) | | 0.1 | | |
| Consistency rampup | 300 | 300 | 300 | 120 |
| Communication rounds | 1000 | 1000 | 1000 | 150 |
| Learning rate $\eta$ | | 0.1 | | |
| Learning rate decay | 0.998 | 0.998 | 0.998 | 0.98 |
| Embedding Size | | | | 512 |
| Number of heads | | | | 8 |
| Dropout | | N/A | | 0.2 |
| Sequence length | | | | 64 |

