# OpenReview forum: "DepthFL : Depthwise Federated Learning for Heterogeneous Clients"
_ICLR.cc/2023/Conference — ICLR 2023 poster_

### Official Review · Reviewer_jAAa · 2022-10-19

**Confidence:** 3
**Correctness:** 3
**Technical Novelty And Significance:** 2
**Empirical Novelty And Significance:** 2
**Recommendation:** 5

**Clarity, Quality, Novelty And Reproducibility:**

Clarity:
- The writing of the experiment part is a little bit confusing and hard to understand. It requires me to give this section multiple passes to kind of grab what the authors want to convey. For example, the terms "global model", "global sub-model", and "sub-model" took me a while to differentiate them.
- Other sections look good to me.

========================================================================

Novelty:
I feel the novelty is kind of limited. There are two main contributions of this paper: 1) propose using depth-wise scaling, and 2) use self-distillation to increase the performance of the model. The first one is the special case of techniques like Federated Dropout and only applicable to networks with branchynet-like architectures, and the second has been widely used to improve network training and the quality of sub-models in one-shot neural architecture search. However, the finding that depth-wise scaling is more beneficial than width-wise scaling under the scope of federated learning is interesting to FL audiences, so I increase the rating for the novelty.

========================================================================

Reproducibility:
Yes.

**Strength And Weaknesses:**

Strength:
- This paper address an important issue of federated learning that many clients are constrained by the available resource and cannot train the whole server model.
- This paper shows that depth-wise scaling is better than width-wise scaling for federated learning.
- This paper shows that self-distillation from the shallower part to the deeper part can help improve the final accuracy.
- Using the almost free model ensemble (under this paper's experimental setting) to enhance the accuracy is clever.
- The experiments are extensive and informative, and I appreciate the authors' efforts in designing and running them.

==============================================================================

Weakness:
- I noticed that in Table 2 the accuracy of exclusive learning of DepthFL (FedAvg) is consistently higher than that of HeteroFL. The sub-model of HeteroFL is thinner but deeper while the sub-model of DepthFL (FedAvg) is shallower but wider. Empirically, the former will perform better than the latter given the same model size. These counter-intuitive results make me wonder whether the gain comes from other sources instead of doing depth-wise scaling, such as more intermediate supervision points (like SHeteroFL) or unfair comparison (as stated in the last sentence of the first paragraph in Sec. 4.2).
- Does HeteroFL use the same architecture as DepthFL with the intermediate supervision points? If not, DepthFL is actually larger and more complicated than HeteroFL.
- Missing the comparison to Federated Dropout. HeteroFL sounds like a special case of Federated Dropout. HeteroFL drops the neurons in a fixed order while Federated Dropout drops the neurons randomly. I wonder whether the results will change if HeteroFL drops different 25% for each client and for each round instead of the same 25%.
- The justification of why depth-wise scaling is better than width-wise scaling in the last paragraph of Sec. 4.2 sounds weird to me. The claim is that the worse accuracy of width-wise scaling is due to the fact that 25% clients train classifier 1/4 directly and 75% clients train classifier 3/4 indirectly. I think it is the same for the proposed depth-wise scaling. For example, the first 25% layers are also trained by 75% clients indirectly due to backprop. Why can this justify the accuracy difference?

Other comments:
Because it uses a branchynet-style network to resolve the resource issue, it requires adding multiple intermediate supervision points with decoders or classifiers. If the decoders and the classifiers are heavy, adding them may nullify the saving in the reduced encoder and increase the complexity of the global model. Some example applications are image reconstruction or speech recognition.

**Summary Of The Paper:**

This paper aims to address the resource challenge of federated learning. Instead of training the same server model across all clients, it proposes scaling the server model along the depth dimension instead of the width dimension to meet the resource limitation of each client. It also found that distilling knowledge from the shallower part to the deeper part is essential to improving the accuracy of the final model ensemble.

**Summary Of The Review:**

Per the discussion in the other sections, I think the novelty of this paper is somewhat limited, and the concerns mentioned in the weakness section make me skeptical about the correctness of the claim that depth-wise scaling is better than width-wise scaling. I will be happy to change my rating if the concerns can be resolved in the rebuttal.

---

> ### Author Response · Authors · 2022-11-14
> **Response**
>
> Thank you for your insightful and helpful comments. We addressed each concern below.
>
> **"I noticed that in Table 2 the accuracy of exclusive learning of DepthFL (FedAvg) is consistently higher than that of HeteroFL. The sub-model of HeteroFL is thinner but deeper while the sub-model of DepthFL (FedAvg) is shallower but wider. Empirically, the former will perform better than the latter given the same model size. These counter-intuitive results make me wonder whether the gain comes from other sources instead of doing depth-wise scaling, such as more intermediate supervision points (like SHeteroFL) or unfair comparison (as stated in the last sentence of the first paragraph in section 4.2)."**
>
> First of all, as shown in Table 1, the sub-model sizes (\# of parameters and \# of MACs) of HeteroFL and DepthFL are quite different. This is the reason why we mentioned in Section 4.2 (at the end of the first paragraph) that we cannot compare the accuracy of DepthFL and HeteroFL directly, and that comparing only with their corresponding exclusive learning is meaningful.
>
> As you noted, exclusive learning of DepthFL(FedAvg) has separate classifiers (intermediate supervision points) in the model, unlike exclusive learning of HeteroFL, and uses the ensemble of internal classifiers during inference, which can lead to better performance. However, we think this is not the point. The point is that the global model of HeteroFL, with 100\% of clients and 100\% of channels, shows a tangibly worse performance than exclusive learning models of HeteroFL (e.g., with 75\% of clients and 50\% of channels for CIFAR-100), due to parameter mismatches when aggregating local models. DepthFL added with those separate classifiers can remove parameter mismatches due to direct supervision of sub-classifiers using companion objectives (see the answer below), thus achieving a better performance than its exclusive learning models. So, even resource-constrained clients can participate and contribute to federated learning if DepthFL is used, unlike HeteroFL.
>
> **"Does HeteroFL use the same architecture as DepthFL with the intermediate supervision points? If not, DepthFL is actually larger and more complicated than HeteroFL."**
>
> You are right. Table 1 shows that the resnet18 global model used in DepthFL has 10\% more parameters and 5\% more MACs due to additional bottleneck layers compared to the plain resnet18 model that HeteroFL uses. Again, this is another reason why we should not compare the performance of DepthFL and HeteroFL directly. On the other hand, this (relatively small) overhead can solve the parameter mismatch problem of HeteroFL and make all clients contribute to federated learning.
>
> **"Missing the comparison to Federated Dropout. HeteroFL sounds like a special case of Federated Dropout. HeteroFL drops the neurons in a fixed order while Federated Dropout drops the neurons randomly. I wonder whether the results will change if HeteroFL drops different 25\% for each client and for each round instead of the same 25\%."**
>
> Actually, both Fjord [1] and HeteroFL use ordered dropout instead of random dropout. This ordered dropout method seems to perform empirically better than random dropout.
>
> [1] FjORD: Fair and accurate federated learning under heterogeneous targets with ordered dropout. NeuRIPS 2021.

---

> > ### Author Response · Authors · 2022-11-14
> > **Response continue**
> >
> > **"The justification of why depth-wise scaling is better than width-wise scaling in the last paragraph of section 4.2 sounds weird to me. The claim is that the worse accuracy of width-wise scaling is due to the fact that 25\% clients train classifier 1/4 directly and 75\% clients train classifier 3/4 indirectly. I think it is the same for the proposed depth-wise scaling. For example, the first 25\% layers are also trained by 75\% clients indirectly due to backprop. Why can this justify the accuracy difference?"**
> >
> > It is actually an important novelty of DepthFL. In DepthFL, when the first 25\% layers (classifier 1/4 weights) are trained by 75\% clients, classifier 1/4 weights are updated in the direction of minimizing both its own loss and the overall loss of its super classifier, using the companion objective and the overall objective, respectively. That is, back-propagation with the overall loss as well as the companion loss will update its hidden layers, as shown in Figure 1 of the revised paper. So, classifier 1/4 is directly supervised. In HeteroFL, however, the channel 1/4 weights are updated in the direction of minimizing the overall loss only, without consideration of the loss of the channel 1/4, so they are computed indirectly.
> > Since the goal of HeteroFL is not to train individual sub-channel, but to train a global model that includes all channels, HeteroFL disregards the performance of a sub-channel. However, as can be seen from the comparison between HeteroFL and SHeteroFL (Table 3), training sub-channels using companion objectives is very important for the overall performance of the global model. In the updated paper, we emphasized this issue more clearly in Section 1 and Section 3.1, and revised Section 4.2.
> >
> > **"Because it uses a branchynet-style network to resolve the resource issue, it requires adding multiple intermediate supervision points with decoders or classifiers. If the decoders and the classifiers are heavy, adding them may nullify the saving in the reduced encoder and increase the complexity of the global model. Some example applications are image reconstruction or speech recognition."**
> >
> > If the size of the decoder is much larger than the size of the encoder and it is impossible to reduce the size of the decoder in any way, this depth scaling method may not be effective. We included this limitation in the revised paper.

---

### Official Review · Reviewer_qicp · 2022-10-24

**Confidence:** 4
**Clarity, Quality, Novelty And Reproducibility:** This work is clearly written with mod…
**Correctness:** 3
**Technical Novelty And Significance:** 3
**Empirical Novelty And Significance:** 3
**Recommendation:** 6

**Strength And Weaknesses:**

Pros:

+ Techically sound approach towards tackling system heterogeneity in FL.
+ Novelty on reverse direction knowledge distillation; authors also provided empirical analysis on its effects.

Cons:
* Extra computation overhead exists in 1) training multiple classifier heads and 2) self-distillation among layers. These classifiers are also required to be transmitted during FL loops.
* Resemblance to early-exit models: The design logic largely reminds me of the early-exit networks (e.g. [1]). Authors are encouraged to cite work along this line and discuss the difference between the proposed work and early exit models.
* Non-comprehensive baselines: authors mentioned different FL approaches of width-based pruning in the related work section, but only FedHetero has been extensively evaluated as the only baseline. When analyzing the lower performance of FedHeteo compared with exclusive learning, I suggest authors study and discuss more related work that enables split learning in FL, such as [2] and [3] which tackled the issue that exists in FedHetero.



[1] BranchyNet: Fast inference via early exiting from deep neural networks.  ICPR 2016.
[2] Fjord: Fair and accurate federated learning under heterogeneous targets with ordered dropout. NIPS 2021.
[3] Resilient and Communication Efficient Learning for Heterogeneous Federated Systems. ICML 2022.


**Summary Of The Paper:**

The authors proposed a new Federated Learning (FL) approach for tackling resource heterogeneity. In their proposed scheme, local models are scaled up horizontally, and FL clients can choose to learn local models with different depths based on their needs. One novelty resides in their design of the reverse direction knowledge distillation, where knowledge is distilled from shallower layers, which were trained by more clients, to deeper layers. As a tradeoff, each layer is attached by a classifier head.

**Summary Of The Review:**

A technically sound work towards addressing system heterogeneity in FL by layer-wise scaling.

---

> ### Author Response · Authors · 2022-11-14
> **Response**
>
> Thank you for the constructive comments. We address each of the concerns below.
>
> **"Extra computation overhead exists in 1) training multiple classifier heads and 2) self-distillation among layers. These classifiers are also required to be transmitted during FL loops."**
>
> You are right since DepthFL requires more parameters and MACs due to additional bottleneck layers, and more computations for self distillation. However, we think that this (relatively small) overhead can solve parameter mismatches when aggregating local models via direct supervision of sub-classifiers, and let all clients participate and contribute to federated learning. We mentioned it in Section 1 of the updated paper as follows: "... This means that depth scaling allows resource-constrained clients to contribute to learning, with a small overhead of separate classifiers."
>
> **"Resemblance to early-exit models: The design logic largely reminds me of the early-exit networks (e.g. [1]). Authors are encouraged to cite work along this line and discuss the difference between the proposed work and early exit models."**
>
> We cited [1,2,3] papers on multi-exit architectures. The model structure of DepthFL is based on [1] paper, and the difference between self distillation in [2,3] and self distillation in DepthFL is discussed in the paper. The [4] paper was additionally cited in the updated paper.
>
> [1] Deeply-Supervised Nets. AISTATS 2015.
> [2] Distillation-based training for multi-exit architectures. ICCV 2019.
> [3] Be your own teacher: Improve the performance of convolutional neural networks via self distillation. ICCV 2019.
> [4] BranchyNet: Fast inference via early exiting from deep neural networks. ICPR 2016.
>
> **"Non-comprehensive baselines: authors mentioned different FL approaches of width-based pruning in the related work section, but only FedHetero has been extensively evaluated as the only baseline. When analyzing the lower performance of FedHetero compared with exclusive learning, I suggest authors study and discuss more related work that enables split learning in FL, such as [2] and [3] which tackled the issue that exists in FedHetero."**
>
> To solve the parameter mismatch, SHeteroFL in Split-Mix, Fjord [2], and FedResCuE [3] learn the prunable local model in slightly different ways. We showed that SHeteroFL solves parameter mismatch through experiments in Section 4.2. Fjord and FedResCuE are slightly different from SHeteroFL, but they have similar characteristics because they use width scaling in common. Since DepthFL uses depth scaling unlike these methods, it has the advantage of obviating the overhead required for learning a prunable local model to eliminate parameter mismatch. Additionally, Appendix A.1 showed the Deeply-Supervised Nets effect of depth scaling while learning a prunable local model.
>
> [2] Fjord: Fair and accurate federated learning under heterogeneous targets with ordered dropout. NIPS 2021.
> [3] Resilient and Communication Efficient Learning for Heterogeneous Federated Systems. ICML 2022.

---

### Official Review · Reviewer_Gy8o · 2022-10-26

**Confidence:** 4
**Clarity, Quality, Novelty And Reproducibility:** 1. It is important to provide a relat…
**Correctness:** 3
**Technical Novelty And Significance:** 2
**Empirical Novelty And Significance:** 3
**Recommendation:** 6

**Strength And Weaknesses:**

**Strengths:**
- The application of depth-wise pruning to tackle resource heterogeneity in FL
- Distillation from shallow to large classifier models via deep mutual learning
- The main ideas in the paper are easy to understand for the reader
- Extensive experiments with ablation studies that help understand the intuition behind design choices


**Weaknesses:**
- Related study is missing discussion on layer-pruning methods
- Novelty of the proposed method appears to be limited as the application of Deeply-Supervised Nets (Lee et al., 2015) and deep mutual learning (Zhang et al., 2018) for FL to improve upon the layer-pruning method (InclusivFL). It might be better to improve Section 1 to clarify the novelty.


**Summary Of The Paper:**

The authors propose Depthwise Federated Learning (DepthFL) framework (includes mutual self-distillation)  to ensure that the global model accuracy improves compared to exclusive FL (excluding resource-constrained clients which can not train global models)) when performing aggregation of local models in FL on heterogeneous clients, especially, resource-constrained devices.
- DepthFL is based on depth scaling in contrast to conventional approaches on width scaling, the latter suffers from degrading accuracy when compared to exclusive FL.
- DepthFL constructs a global model that has several classifiers of different depths. It prunes the highest-level layers of the global model to create local models with different depths based on the client's available resources, thus with a different number of classifiers.
- Each client not only trains the classifiers in its local model using local data but also distills the knowledge  across classifiers at the same time
-  For inference, DepthFL uses the ensemble of all internal classifiers while HeteroFL uses the global model with all channels

**Summary Of The Review:**

This paper tackles an important problem of FL on heterogeneous clients. The idea is simple to understand which is to have clients with different resource availabilities have local models of varying capacity ( here, a depth-wise pruned version of the global model). However, the paper is written in a way that shows it as a limited novelty by applying Deeply-Supervised Nets (Lee et al., 2015) and deep mutual learning (Zhang et al., 2018). Moreover, an important extensive comparison with InclusiveFL is missing from the main body (though a section exists in Appendix) which is closest in design to DepthFL and has the same baseline comparisons used in this work.

However, I will be willing to update my score based on overall discussion and clarifications of the issues raised.

---
Edit: I have read the authors' responses and increased my score.

---

> ### Author Response · Authors · 2022-11-14
> **Response**
>
> Thank you for your insightful comments. We address the comments as below.
>
> **"Related study is missing discussion on layer-pruning methods"**
>
> In Section 2 of the revised paper, we added related studies on layer-pruning methods in addition to InclusiveFL, and on multi-exit architectures.
>
> **"Novelty of the proposed method appears to be limited as the application of Deeply-Supervised Nets (Lee et al., 2015) and deep mutual learning (Zhang et al., 2018) for FL to improve upon the layer-pruning method (InclusiveFL). It might be better to improve section 1 to clarify the novelty."**
>
> In Section 1 of the revised paper, we clarified the novelty of DepthFL by adding a comparison with InclusiveFL as follows:
>
> "Recently, InclusiveFL (Liu et al., 2022) proposed a kind of depth-scaled method with a similar intuition to ours, yet with less elaboration. We show that depth scaling alone in InclusiveFL without the companion objective of sub-classifiers cannot effectively solve parameter mismatches, and that its performance of deep classifiers is much lower due to no self distillation."
>
> Actually, the novelty of DepthFL that employs Deeply-Supervised Nets (DSN) is not just for depth-based pruning but for solving parameter mismatches when local models are aggregated (which is a problem of width-based pruning), by exploiting DSN's integrated direct supervision to sub-classifiers. We added the following statement in Section 3.1:
>
> "DSN can provide integrated direct supervision to a sub-classifier in addition to the output classifier, since back-propagation with the overall loss as well as the companion loss will update its hidden layers, as shown in Figure 1. DSN exploits it for better performance, but DepthFL exploits it for consistent training of a sub-classifier across all local models that includes it to fully exhibit its performance."
>
> We also revised the first contribution item in Section 1, as follows:
>
> "We present a new depth scaling method to create heterogeneous local models with sub-classifiers, which directly supervises sub-classifiers, obviating their parameter mismatch when aggregated."
>
> **"It is recommended to provide the comparison with a layer-pruning method (InclusivFL) in the main body as one of the baseline comparisons, more so, because InclusiveFL uses the same baselines such as Exclusive FL and HeteroFL"**
>
> We moved some of the comparative experiments and explanations on InclusiveFL from the appendix to Table 3 and Section 4.2 of the revised paper.
>
> **"Fig. 1 should be redone to improve the quality and illustration.''**
>
> We updated Figure 1 to express the characteristics of DepthFL more clearly, by showing the companion loss, the distillation loss, and the overall loss, and by comparing the capacity and the training data amount for each layer.
>
> **"It is unclear if the experiments used 100 units of Nvidia RTX 2080 Ti GPUs as FL clients or simulations. More discussion on FL setup is required."**
>
> We simulated by having several worker processes occupying single GPU each, and distributing each client's local training workload to the worker processes in every communication round.

---

### Official Review · Reviewer_nCtb · 2022-10-31

**Confidence:** 3
**Correctness:** 4
**Technical Novelty And Significance:** 4
**Empirical Novelty And Significance:** 4
**Recommendation:** 8

**Clarity, Quality, Novelty And Reproducibility:**

I found the paper quite easy to read, the evaluation section was good quality, and overall work is quite novel.

**Strength And Weaknesses:**

The paper addresses an important problem by enabling resource-constrained clients in heterogeneous FL. I found the two techniques of varying depths to match client resources and self-distillation of the deeper layers to be quite intuitive. I also found the evaluation section to be well organized, motivated, and presented.

The only weakness is that I would have liked to see an evaluation of a scenario where 100% of the clients could participate in training the global model, as that could help to understand the upper bound performance of these tasks.

**Summary Of The Paper:**

The paper proposes DepthFL, an approach that enables resource-constrained clients to participate in Federated Learning (FL) of a global model that is larger than these devices. DepthFL defines shallower versions of the global model for simultaneously training, whereby each client trains a local model of a depth appropriate for its resources.  DepthFL employs self-distillation of the highly trained shallow layers to address the insufficient training of deeper layers. The experimental results show improved results for both the global model and the shallow local models compared to excluding resource-constrained devices.

**Summary Of The Review:**

The paper convinced me that local models of varying depths and self-distillation is an effective way to improve performance and usefulness of FL on heterogeneous clients.

---

> ### Author Response · Authors · 2022-11-14
> **Response**
>
> Thank you for your helpful comments. We address the comments as below.
>
> **"The only weakness is that I would have liked to see an evaluation of a scenario where 100\% of the clients could participate in training the global model, as that could help to understand the upper bound performance of these tasks."**
>
> The **Maximum** scenario in Section 4.3 is when all clients can participate in training of the full global model. The evaluation result of the **Maximum** is shown in Table 5, which constitutes the upper bound for other scenarios such as **Fixed** or **Dynamic**.

---

### Public Comment · ~Rivan_Ilk1 · 2022-11-19
**To reviewers and authors, regarding limited comparisons and evaluations of the work**

Dear authors and reviewers,

The paper is enjoyable to read. The reviewers’ comments and the authors’ responses are quite insightful as well. In this regard, the following points may be helpful for further evaluation of the current work.

**Missing important comparisons with related work**

WIth similar intention as that of DepthFL, InclusiveFL[1] and FjORD[2] also tried to solve device heterogeneity. Moreover, both FjORD and inclusiveFL used distillation to improve the performance under such settings. Interestingly, InclusiveFL also leveraged the depth heterogeneity as this work to address device heterogeneity. To understand the true value of the paper, a fair and possibly more comprehensive comparison with these works is expected.  For example, it is not clear from Table 3 whether the accuracies are compared on a similar parameter and/or FLOPs budget.

**Concern: additional compute and storage cost associated to BottleNeck blocks of Auxiliary classifiers**

The original self-distillation paper [3] used parameter heavy bottleneck layers (auxiliary classifier modules), particularly for the earlier ResNet blocks, use of similar sized auxiliary classifier modules would diminish the computation and storage advantage in case of FL. Moreover, as the authors assume clients 1,2 (in Fig. 1) are of lower capacity, application of these auxiliary modules often may not be feasible due to the significant additional overhead. In contrast FjORD can leverage width dropping (instead of depth dropping), without essentially incurring any need of auxiliary classifier module.

**Potentially unfair discussion on Exclusive Learning**

It seems that the motivation and discussion provided in Section 4.2 is a bit unfair. First of all, the authors should use FjORD [2] instead of HeteroFL [5] for their comparisons. Secondly, the authors have considered the simplistic IID setting where missing data from a subset of clients may not be a major concern. However, in the practical scenarios of extreme non-IID such as [4], some clients may have exclusive access to a subset of training labels. In such cases, ignoring updates from those clients (as done in Exclusive Learning) would clearly degrade training performance as the model never sees updates belonging to that exclusive subset of labels. On the other hand, HeteroFL/FjORD would be able to still observe those updates. It will be of significant interest to the wide FL community if the authors provide a similar discussion (as in Section 4.2) for extreme non-IID settings.

[1] Ruixuan Liu, Fangzhao Wu, et al. No one left behind: Inclusive federated learning over heterogeneous devices. In KDD, 2022.
[2] Horvath, S., Laskaridis, S., et al. FjORD: Fair and accurate federated learning under heterogeneous targets with ordered dropout. In NeurIPS, 2021.
[3] Zhang, L., Song, J., Gao, A., Chen, J., Bao, C. and Ma, K. Be your own teacher: Improve the performance of convolutional neural networks via self distillation. In ICCV, 2019.
[4] McMahan, B., Moore, E., Ramage, D., Hampson, S. and y Arcas, B.A. Communication-efficient learning of deep networks from decentralized data. In AISTATS, 2017.
[5] Diao, E., Ding, J. and Tarokh, V. HeteroFL: Computation and Communication Efficient Federated Learning for Heterogeneous Clients. In ICLR, 2020.

---

> ### Author Response · Authors · 2022-11-21
> **Response**
>
> Dear Rivan,
>
> Thank you very much for your time to read our paper and provide insightful comments. Here are the answers to your comments.
>
> **"Potentially unfair discussion on Exclusive Learning"**
>
> The comparison experiment with exclusive learning in Sec 4.2 is to identify the parameter mismatch problem, not to show that exclusive learning is superior to the corresponding algorithm. Through comparative experiments with exclusive learning, we could deduce that parameter mismatches are decisive for performance, and that direct supervision with companion objective is important.
>
> **"Concern: additional compute and storage cost associated to Bottleneck blocks of Auxiliary classifiers"**
>
> First of all, the sizes of local models in Table 1 include all bottleneck layers. It shows that the average communication overhead of depth-scaled clients would be lower than that of width-scaled clients, but it would be opposite for the average computation overhead. Despite the overhead of the bottleneck layers, depth scaled local models have fewer parameters than width scaled local models because most of the parameters are concentrated in the deep layers. In fact, the number of parameters of the shallow local models, excluding the parameters of the bottleneck layers, is much smaller than that recorded in Table 1.
>
> **"Missing important comparisons with related work"**
>
> DepthFL has two novelties: (1) depth-based pruning with direct supervison of sub-classifiers, (2) self distillation (w/ reverse direction knowledge distillation). First of all, InclusiveFL performs depth-based pruning but no direct supervison of sub-classifiers, and the InclusiveFL result in Table 3 was obtained as such for the same global model of DepthFL(FedAvg); so the only difference between DepthFL(w/o self distillation) and InclusiveFL(w/o momentum distillation) in Table 3 is whether there is direct supervison of sub-classifiers or not (so there is a overhead of sub-classifier for DepthFL(FedAvg)). Since there is no direct supervison of sub-classifiers, InclusiveFL appears to suffer from parameter mismatches when aggregated, exactly as noted by the last reviewer jAAa (“…I think it is the same for the proposed depth-wise scaling. For example, the first 25\% layers are also trained by 75\% clients indirectly due to backprop.”).
>
> To solve the parameter mismatch, SHeteroFL in Split-Mix[1], Fjord [2], and FedResCuE [3] learn the prunable local model in slightly different ways. We showed that SHeteroFL solves parameter mismatch through experiments in Section 4.2. Fjord and FedResCuE are slightly different from SHeteroFL, but they have similar characteristics because they use width scaling in common. Since DepthFL uses depth scaling unlike these methods, it has the advantage of obviating the overhead required for learning a prunable local model to eliminate parameter mismatch. Additionally, Appendix A.1 showed the Deeply-Supervised Nets effect of depth scaling while learning a prunable local model.
>
> [1] Efficient split-mix federated learning for on-demand and in-situ customization. ICLR 2022.
> [2] Fjord: Fair and accurate federated learning under heterogeneous targets with ordered dropout.
> NIPS 2021. [3] Resilient and Communication Efficient Learning for Heterogeneous Federated Sys-
> tems. ICML 2022.

---

### Decision · Program_Chairs · 2023-01-20

**Decision:**

Accept: poster

**Justification For Why Not Higher Score:**

- Limited novelty and position in the existing relevant literature: Several reviewers have expressed concerns about limited novelty as the proposed method is an application of Deeply-Supervised Nets (Lee et al., 2015) and deep mutual learning (Zhang et al., 2018) for FL to improve upon the layer-pruning method (InclusivFL). Related work is missing discussion on layer-pruning methods. Reviewers suggest more extensive discussions on comparison with early-exit models.

- Extra computation overhead exists in 1) training multiple classifier heads and 2) self-distillation among layers. These classifiers are also required to be transmitted during FL loops. Since the proposed method uses a branchynet-style network to resolve the resource issue, it requires adding multiple intermediate supervision points with decoders or classifiers. If the decoders and the classifiers are heavy, adding them may nullify the saving in the reduced encoder and increase the complexity of the global model.

- Reviewers have also suggested more baselines for comparison.

**Justification For Why Not Lower Score:**

This paper considers an important and timely problem. It has strong
technical contribution, using a self-distillation from the shallower part
to the deeper part to improve the final accuracy. The paper also
demonstrates strong empirical evaluation with experiments designed and
conducted extensively.

**Metareview: Summary, Strengths And Weaknesses:**


This paper addresses an important issue of federated learning that many clients are constrained by the available resource and cannot train the whole server model. The authors propose Depthwise Federated Learning (DepthFL) framework together with mutual self-distillation to do depth scaling.  DepthFL constructs a global model that has several classifiers of different depths, and prunes the highest-level layers of the global model to create local models with different depths based on the client's available resources. Each client not only trains the classifiers in its local model using local data but also distills the knowledge across classifiers at the same time.


Strengths:
+ Relevance of the topic: the problem considered is important and timely.

+ Technical contribution: the paper uses a self-distillation from the shallower part to the deeper part to improve the final accuracy.

+ Strong empirical evaluation: experiments designed and conducted are extensive and informative.

Weaknesses:

- Limited novelty and position in the existing relevant literature: Several reviewers have expressed concerns about limited novelty as the proposed method is an application of Deeply-Supervised Nets (Lee et al., 2015) and deep mutual learning (Zhang et al., 2018) for FL to improve upon the layer-pruning method (InclusivFL). Related work is missing discussion on layer-pruning methods. Reviewers suggest more extensive discussions on comparison with early-exit models.

- Extra computation overhead exists in 1) training multiple classifier heads and 2) self-distillation among layers. These classifiers are also required to be transmitted during FL loops. Since the proposed method uses a branchynet-style network to resolve the resource issue, it requires adding multiple intermediate supervision points with decoders or classifiers. If the decoders and the classifiers are heavy, adding them may nullify the saving in the reduced encoder and increase the complexity of the global model.

- Reviewers have also suggested more baselines for comparison.

Although there are some concerns regarding limited novelty, the merits of the paper, especially its strong empirical evaluation, outweigh.

**Note From Pc:**

if the above contains the word "oral" or "spotlight" please see: "oral" presentation means -> notable-top-5% and "spotlight" means -> notable-top-25%. As stated in our emails, we are disassociating presentation type from AC recommendations